# Regional circulation patterns of Mediterranean Outflow Water near the Iberian and African continental slopes

Álvaro de Pascual-Collar[1,2], Marcos García-Sotillo[1], Bruno Levier[3], Roland Aznar[1,2], Pablo Lorente[1,2], Arancha Amo-Valadrón[1,2], Enrique Álvarez-Fanjul[1]

[1]Puertos del Estado, Avenida del Partenón 10, 28042, Madrid, Spain.
[2]Nologing, Avda. De Ranillas 1D, 50018 Zaragoza, Spain.
[3]Mercator Ocean, 8-10 Rue Hermès, 31520 Ramonville Saint-Agne, France.

*Correspondence to*: Álvaro de Pascual (alpascua@ucm.es)

**Abstract.** The Mediterranean Outflow Water (MOW) is a dense water mass originated in the Strait of Gibraltar. Downstream of the Gulf of Cadiz, the MOW forms a reservoir region west of the Iberian continental slopes at a buoyant depth of approximately 1000 m. This region plays a key role as the main centre where the MOW is mixed and distributed into the North Atlantic. The seafloor in this area is characterised by the presence of a complex bathymetry with three abyssal plains separated by mountain chains. Despite the topographic features do not reach the surface, they influence ocean flows at intermediate and deep ocean layers conditioning the distribution and circulation of MOW.

The CMEMS IBI ocean reanalysis is used to provide a detailed view of the circulation and mixing processes of MOW near the Iberian and African Continental slopes. This work emphasizes the relevance of the complex bathymetric features defining the circulation processes of MOW in this region. The high resolution of the IBI reanalysis allows to make a description of the meso-scale features forced by the topography. The temperature, salinity, velocity, transport, and vorticity fields are analysed to understand the circulation patterns of MOW. The high-resolution circulation patterns found reveal that Horseshoe Basin and the continental slope near Cape Ghir are key areas controlling the mixing processes of MOW with the surrounding waters masses, mainly North Atlantic Central Water (NACW), and Antarctic Intermediate Water (AAIW). The water masses variability is also analysed by means of composite analysis. Results indicate the existence of a variability of the MOW tongue which retracts and expands westwards in opposition to the movement of the underlying North Atlantic Deep Water.

## 1 Introduction

The Mediterranean Outflow Water (MOW) is a saline and warm water mass principally occupying the intermediate depths of eastern North Atlantic. It is generated by the outflow of subsurface Mediterranean Sea Water (MSW) through the Strait of Gibraltar. After exiting the Strait of Gibraltar, the denser MSW cascades along the slope in the Gulf of Cadiz and progressively entrains the ambient North Atlantic Central Water (NACW). Its downward flow is disrupted by several transverse submarine valleys splitting up the stream in two different branches of different densities. Both branches flow along different erosive channels, but finally converge near Cape St. Vincent forming on the way two differenced layers of the same water mass (Iorga

and Lozier, 1999a and 1999b, Gasser et al., 2017). MOW exits the Gulf of Cadiz at a buoyant depth around 1000 m and afterward it turns northwards following the western margin of the Iberian Peninsula. It enters the Tagus Basin where it turns anticyclonically forming a reservoir of this water mass (Daniault et al., 1994). From this reservoir zone, the MOW spreads into the North Atlantic, following two main advective pathways: the westward branch towards the central Atlantic and the poleward

branch that, driven by the Iberian Poleward Current, follows the western European continental slope. In its way north MOW enters the Bay of Biscay and continues further north towards Porcupine Bank and Rockall Trough, at 53ºN (Reid, 1979, 1994; van Aken and Becker, 1996; Iorga and Lozier, 1999a, 1999b; Bower et al., 2002). Downstream of the poleward pathway, diapycnal mixing occurs with the underlying Labrador Sea Water core (LSW). It is one of the recognized ways of salt transport into the inner North Atlantic (Talley and McCarney, 1982; van Aken, 2000). The influence of MOW on the Nordic seas has

been discussed in several studies (Reid, 1979, 1994; Bower et al., 2002; Iorga and Lozier, 1999a, 1999b; New et al., 2001; McCarney and Mauritzen, 2001), however later studies suggest that westward shifts of the subpolar front controls the input of Atlantic water through Rockall Trough (Lozier and Stewart, 2008).

In addition to these two main paths, the MOW spreads from the reservoir in south and south-west direction under the influence of other processes and interacts with surrounding water masses. Thus, another core of MOW spreads south-westwards largely

supported by the motion of Mediterranean Water lenses, so-called meddies (Armi and Zenk, 1984; Armi et al., 1989; Bower et al., 1997; van Aken, 2000). The saline signal of MOW extends southward, reaching Canary Islands where the water mass collides with a diluted form of Antarctic Intermediate Water (AAIW) (Machín et al. 2009; Machín and Pelegí 2010).

Several works have focused on the description of MOW in the Gulf of Cadiz, some of them based on observational in-situ data (García-Lafuente et al., 2006; Machín et al., 2009) and others based on model products (Artale et al., 2006; Gasser et al., 2017;

Izquierdo and Mikolajewicz, 2019). The consistency between both approaches has improved the knowledge of the oceanographic processes taking place in this area. However, the MOW flow beyond the Gulf of Cadiz has not been investigated as extensively and the scientific literature is rather limited. Most of the studies in this area are based in compiling data sets of hydrographic stations or climatological data (e.g. Mazé et al., 1997; Iorga and Lozier 1999a; van Aken, 2000; Carracedo et al., 2014). Despite the fact these methodologies consider a fair number of stations, they have a limited spatio-temporal coverage,

therefore the results may be influenced by seasonal or mesoscale processes. On the other hand, the studies based on numerical simulations are usually focused on larger scales such as the eastern North Atlantic basin scale (e.g. Iorga and Lozier, 1999b; Bozec et al., 2011) with a resolution too coarse to provide a proper representation of mesoscale processes at a regional scale. However, several scientific studies have described the existence of important mesoscale features in this region such as: formation of Mediterranean Water lenses (meddies; Armi and Zenk, 1984; Armi et al., 1989; Bower et al., 1997; Daniault et

al., 1994; Sangrá et al., 2009), development of an anticyclonic gyre centred in the Tagus Abyssal Plain, and flow along the Gorringe Bank; all highlighting the influence of bathymetry on the development of mesoscale flows (Zenk and Armi, 1990; Daniault, 1994; Iorga and Lozier, 1999a, and 1999b). Therefore, we have a good understanding of the general circulation of MOW, but there is still a lack of knowledge about the high resolution circulation patterns affecting this water mass along its way.

Ocean reanalysis provide nowadays consistent and realistic 3D gridded ocean fields over the last decades. The combination of numerical ocean modeling and data assimilation techniques allows to get a realistic view of the circulation patterns including: (1) high-enough spatial resolution to reconstruct the mesoscale features and (2) long temporal coverage allowing to compute averaged fields, thus filtering the temporal variability and retaining stationary features. The present work uses the Copernicus

Marine Environmental Monitoring Service (CMEMS) Iberian-Biscay-Ireland (IBI) reanalysis (Levier et al., 2014, Sotillo et al., 2015, Aznar, 2016, Amo Baladrón, 2018) to make a description of the mean circulation in a domain encompassing the Tagus, Horseshoe and Seine abyssal plains. It emphasizes the mesoscale features together with the general circulation patterns and their, interannual variability. The CMEMS IBI reanalysis has been produced by the CMEMS IBI Monitoring and Forecasting Centre (IBI-MFC) and, it is one of the muti-year data product used in the Copernicus Ocean State Report (von

Schuckman et al. 2017, von Schuckman et al., 2018) which is aimed to provide regular and systematic reference information on the physical state, variability and dynamics of the global ocean and European regional seas. In the framework of its second issue, the IBI-MFC proposed an ocean monitoring indicator to characterize the interannual variability of MOW in the IBI region (Pascual et al., 2018); to that purpose, an analysis of the circulation patterns, transports, and temporal variability of MOW was performed. The present work summarize part of the results from in the analysis done specifically for the design

phase of the CMEMS MOW Ocean Indicator. The resolution of the IBI reanalysis (1/12º, ~8 Km) allows to observe several circulation features that reveal the important role of bathymetry in the spreading of MOW beyond the Gulf of Cadiz. Additionally, the long temporal coverage of the IBI reanalysis (1992-2016) permitted the analysis of the mid-term variability of MOW in the region.

The paper is organized as follow: a preliminar background of the distribution of MOW near the African and Iberian continental

slopes is given in section 2. Description of the IBI reanalysis and the methods used in the following sections are presented in section 3. Results are presented and discussed in sections 4 and 5: the analysis of flow, transports and hydrographic properties is split in two subsections focused in Tagus and Horseshoe basins (section 4.1) and the other on the African slope near Cape Ghir and Cape Sim (section 4.2); the temporal variability of MOW is studied in section 5. Finally, main conclusions are summarized in section 6.

**2 Background on the MOW regional behaviour**

The region of study is bounded by the 39ºN and 31ºN parallels, and by the 15ºW meridian on the West and the Iberian and north-west African continental shelf edges on the East (Figure 1). This domain comprises three abyssal plains oriented from North to South (Tagus, Horseshoe, and Seine) with depths greater than 5000 m. These three abyssal plains are separated by a number of seamounts structured in two zonal mountain chains, whose highest peaks are Ampere Bank and Gorringe Bank.

The west margin of the plains is limited by the presence of the Azores-Portugal Rise. These promontories are quite high obstacles that clearly separate the three basins, which are delimited by a set of walls around each plain. Several works on the MOW distribution have concluded that the complex orography found in this region affects the MOW flow. The outflow MOW

from the Gulf of Cadiz is strongly affected by the existence of a narrow passage between Gorringe Bank and Cape St. Vincent (Figure 1). Zenk and Armi (1990) proposed a schematic flow pattern for the MOW after its passage through this "gateway". Later studies have concluded that, despite of most of the MOW flowing out of the Gulf of Cadiz turns northward through this narrow passage (Daniault et al., 1994), a second branch of the flow is deflected westwards along the southern flank of the Gorringe Bank (Iorga and Lozier, 1999a and 1999b).

Another feature commonly described in works is the formation of two gyres. The first one, a cyclonic gyre, attached to the Gulf of Cadiz centred at ~35ºN and ~9ºW with a diameter of ~250 Km (Iorga and Lozier, 1999a, and 1999b), and the second one, is an anticyclonic gyre centred in the Tagus Basin. The latter is the main responsible of the accumulation area known as MOW reservoir (Iorga and lozier, 1999a and 1999b). It is worth mentioning that the definition of the MOW reservoir has been widely used in several studies, however the geographic window used to define this area may differ considerably from the more restrictive definitions (e.g. Iorga and Lozier, 1999a; and Pascual et al. 2018) to the more relaxed ones (e.g. Potter and Lozier, 2004; Bozec, 2011).

The hydrographic properties of MOW that reside in the reservoir are considerably different than the ones of the original MSW generated in the Strait of Gibraltar. This is due to the intense mixing processes taking place in the Gulf of Cadiz, which mainly imply the downwelling and entrainment of NACW (Jia, 2000; Carracedo et al. 2014), as well as of other ambient water masses. As the reservoir is mostly composed by the mixing of MSW and NACD, a possible source for the variability of MOW properties in the reservoir could be the variability of MSW or/and NACW (Artale et al., 2006, Fucso et al., 2008). However, other studies (Baringer and Price, 1997; Lozier and Sindlinger, 2009) stated that these two possibilities are too weak to explain the observed variability of MOW in the reservoir. In a more recent study, Bozec et al. (2011) concluded that MOW reservoir variability can be explained by changes of the North Atlantic circulation resulting in a shift of the preferred MOW pathway, such changes being induced by the variability of the atmospheric forcing.

The area covered by the Tagus anticyclonic gyre is a key location from where the MOW is distributed on its two main advective pathways. North of the Tagus gyre, part of the flow diverges and continues northward beyond the Extremadura Promontory. The southern boundary of the Tagus gyre is linked with the westward MOW branch defined by Reid (1994), This branch originates in the southern flank of Gorringe Bank, and is fed by the splitting of the flow leaving the Gulf of Cadiz. It travels westwards along the northern wall of Horseshoe Basin (Iorga and Lozier, 1999a) where it converges with the southward geostrophic flow coming from the Tagus gyre (Iorga 1999a, Daniault et al., 1994). The formation of the so-called westward pathway geographically starts when the flow detaches from the mountain chain and penetrates westwards into the North Atlantic. However, Iorga and Lozier (1999a and 1999b), using 80 years of hydrographic data and a diagnostic model, found that this branch does not count as a significant input of MOW into the subtropical gyre.

Some evidences suggest that there is some advective southward flow along the African continental slope. It is originated in the cyclonic gyre centred in the Gulf of Cadiz spreads southwards along the African continental slope (Iorga and Lozier, 1999a) and has been observed at latitudes of 32ºN (Machín and Pelegrí, 2009). However, later studies (Izquierdo and Mikolajewicz, 2019) have stated that the tidal forcing play a major role limiting the southward advective transport of MOW and contributing

to the advection of MOW west from the Gulf of Cadiz. The general scientific consensus on the spread of MOW south of Gulf of Cadiz points out that it is largely supported by the development of meddies created southwest of the MOW reservoir and near the African continental slope (Zenk et al., 1992; Sangrá et al., 2009). As studies of the penetration of the westward branch into the North Atlantic does not show a clear advective transport of MOW into the subtropical gyre, the formation of meddies is considered the main cause of the observed large-scale westward penetration of Mediterranean salt (Arham and King, 1995). The southern boundary of the MOW is affected by the interaction with the overlying AAIW, explaining the deeper salinity signature of MOW at these latitudes (~1200 m) due to the presence of the AAIW core at about 800 m depth. According to Machín and Pelegrí (2009) the northern limit of AAIW is located around 32ºN near the African continental slope, however their presence northward of the Canary Islands is subjected to seasonal variability. The mixing processes between both water masses have been studied by van Aken (2000); they conclude that the core of AAIW appears to contribute to the formation of MOW since it is entrained into the overflow near Gibraltar. Such entrainment gives rise to an enhanced concentration of the nutrients in the Mediterranean originated water in the North Atlantic.

Several studies have focused on the temporal variability of the MOW and have concluded that their variability is within a wide range of time scales. Prieto et al. (2013) analysed a dataset of semiannual hydrographic stations located on the Iberian platform. Their findings point out that despite a larger interannual variability, the seasonal signature represents 20% of the interannual variability. This variability has been found along the continental slopes affected by MOW, and is a consequence of the variability of the slope current responsible of the northwards transport of MOW along the European continental slopes. In summertime when the narrow jet is trapped along the continental slope, its thermohaline signature is reinforced near the slope and diminished thereby its presence in the open ocean. In wintertime the slope current weakens and can even be reversed in some areas (Fricourt et al., 2007; Prieto et al., 2013; Pascual et al., 2018).

Variability at longer time scales has been also addressed by several studies. Potter and Lozier (2004) analysed 40 years of hydrographic data to calculate temperature and salinity trends of the MOW reservoir. Along this period, they found positive temperature and salinity trends that lead to a heat content gain that overpasses the average gain of the North Atlantic basin over the later half of the 20th century. Leadbetter et al. (2007) using three repeated sections at 36ºN in the North Atlantic, found an increase of MOW salinity from 1959 to 1981 followed by an almost compensated decrease in salinity from 1981 until 2005 in the upper-intermediate layers. According to their results, this change is controlled by water mass changes along neutral density surfaces suggesting a change in the source waters. Bozec et al. (2011) hypothesized that MOW water mass distribution may be altered by changes in the circulation of the North Atlantic: to investigate this possible source of variability they used a set of model runs forced by either a climatological forcing or an interannual atmospheric forcing. They found a connection between salinity anomalies along the northern and the westward pathways, thereby concluding that the observed salinity changes in the MOW reservoir can be explained by circulation-induced shifts in the salinity field in eastern North Atlantic basin.

# 3 Data and methods

The present study uses the IBI ocean reanalysis delivered by CMEMS (Levier et al., 2014; Sotillo et al. 2015; Aznar et al., 2016, Amo Baladrón, 2018). The domain covered by the IBI reanalysis is limited by the 26ºN and 56ºN parallels, and the 19ºW and 5ºE meridians. It provides daily averages of zonal and meridional velocity components for a period that ranges from
January 1992 to December 2016, with a 1/12º horizontal resolution and 75 vertical levels.

The numerical core of the IBI reanalysis is version 3.6 of eddy-permitting NEMO ocean general circulation model (Madec, 2008). This model solves the three-dimensional finite-difference primitive equations in spherical coordinates discretized on an Arakawa-C grid. It assumes hydrostatic equilibrium and Bousinesq approximation. Vertical mixing is parameterized according to a k-ε model implemented in the generic form proposed by Umlauf and Burchard (2003) including surface wave breaking
induced mixing. The bathymetry is derived from GEBCO 08 dataset (Becker et al., 2009), merged with several local databases. The IBI run is forced with atmospheric fields from the ECMWF ERA Interim (Dee et al., 2011). 10-m wind, surface pressure (added with inverse barometer approximation), 2-m temperature and relative humidity are provided at 3-hour frequency. On the contrary, precipitation and radiative fluxes are provided as daily averages from a modified ERA-interim reanalysis (Sotillo et al., 2015). CORE empirical bulk formulae (Large and Yeager, 2004) are used to compute latent sensible heat fluxes,
evaporation and surface stress.

The IBI reanalysis uses lateral open boundary data from the CMEMS global reanalysis (temperature, salinity, velocities and sea level) at a resolution of 0.25º (Garric and Parent, 2018). These are complemented by eleven tidal harmonics built from FES2004 (Lyard et al., 2006) and TPXO7.1 (Egbert and Erofeeva, 2002). Fresh water river discharge is implemented as lateral
open boundary condition for 33 rivers from observational and climatological data. Additionally an extra coastal runoff rate climatology is used to make the IBI forcing consistent with the ones imposed in the global system (Maraldi et al. 2013).

The data assimilation scheme applied is the MERCATOR Ocean SAM2 (Lellouche et al., 2013), established from a Singular Extended Evolutive Kalman filter. Measurements from CMEMS CORA product (Cabanes et al., 2013; Gatti and Pouliquen, 2017) database are assimilated as well as high resolution Sea Surface Temperature data obtained from analysis of multi-satellite
and AVHRR products, and remote sensing Sea Level Anomalies (SLA) measured by radar altimeter (Jason-3, Sentinel-3A, HY-2A, Saral/AltiKa, Cryosat-2, Jason-2, Jason-1, T/P, ENVISAT, GFO, ERS1/2).

This study is based on the regional analysis of the IBI averaged circulation between 500 and 1500 m for the period 1992-2016. IBI reanalysis provides ocean fields at 947 and 1045 m depth, therefore in this work the 1000 m depth fields shown are derived from the average of this two IBI levels. The resulting average circulation field offers information about the main transports
occurring in the region. Since the time average is computed from daily data at 1/12º resolution, the analysed fields include the net mesoscale transports filtering its intrinsic temporal variability. The interannual variability of MOW and its associated oceanic patterns is studied by the analysis of the high/low salinity events in Horseshoe Basin at 500-1500 m depth. These

events are defined as the $10^{th}$ and $90^{th}$ percentiles of the salinity time series. The composites of temperature and salinity are derived from these events providing an image of the average ocean state under these conditions.

Part of the analysis is focused over several specific areas (boxes and sections defined in Figure 1). These regional domains have been selected taking into account general circulation patterns. Horseshoe basin is split in two different boxes: The northern one (Gorringe Bank Box, GBB) comprises Gorringe Bank and the adjacent western mountain chain, the southern box (Ampere Bank Box, ABB) surrounds the Ampere Bank and the seamounts that form the southern boundary of Horseshoe Basin. The third box is defined north of the Canary Islands near to Cape Ghir (Cape Ghir Box, CGB). Its limits were established surrounding the bathymetric promontory located in the African continental slope between Cape Ghir and Cape Sim. Additionally, a meridional section south of Cape St. Vincent (Cape St. Vincent Section, CVS) has been defined to compute the net transport and water properties flowing out of the Gulf of Cadiz. The analysis of water properties, circulation, and transports is made by averaging over each box (from 500 to 1500 m), as well as along the sections limiting the boxes and CVS. The volume transports shown are computed between 500 and 1500 m considering the vertical surface of each grid cell. The confidence interval of transports has been estimated by bootstrapping techniques.

## 4 MOW regional circulation patterns

### 4.1 Tagus and Horseshoe Basins

Figure 2 shows the mean 1000 m velocity field derived from IBI Reanalysis. MOW transport along the northern slope of Gulf of Cadiz appears as narrow stream that flows westward along the slope. The averaged velocity in this region can reach 0.5-1 m/s, a value consistent with the current velocities found in the literature (Gasser et al., 2017; Sánchez-Leal et al., 2017). South of the outflow stream, results suggest the existence of an eastward countercurrent that provides the external water input required to feed the intense entrainment processes generated by MOW in the region (Ambar and Howe, 1979; Ochoa and Bray, 1991; Baringer and Price, 1997 ; Jia, 2000).

Beyond Cape St. Vincent the flow is strongly affected by the orography. After the MOW exits the Gulf of Cadiz, the main stream splits when encountering Gorringe Bank. One part of the flow turns northward through the narrow passage between Cape St. Vincent and Gorringe Bank and the other part continues westward south of this promontory. The northward path develops an intense stream that follows the Portuguese continental slope and enters the Tagus Basin. After the flow crosses the gateway, it diverges again to form two separated circulation areas: the bathymetric channelling of water towards the Tagus Basin that leads to the formation of the widely-described Tagus Anticyclonic Gyre; and a narrow northward flow that, despite of the orographic elevations, goes northward following the margin defined by the continental slope.

The area formed by the Tagus Basin and Gorringe Bank can be seen as the centre where the MOW is distributed into the North Atlantic. The narrow passage between Gorringe Bank and Cape St. Vincent forces the splitting of the flow exiting the Gulf of Cadiz and promotes the formation of the two main advective pathways of MOW. The westward MOW pathway begins when the water masses detach from the continental slope south and north of Gorringe Bank and travel following the chain of

seamounts west of this promontory. On the southern flank of the mountain range, the flow is forced by the limited transport through the Gorringe gateway; on the northern flank of Gorringe Bank, the westward flow follows the southern limit of the Tagus Anticyclonic Gyre. The northward branch of MOW commences after the MOW enters the Tagus Basin. Here the flow spits again into two separated features: the Tagus Anticyclonic Gyre, and the poleward slope current which starts near a promontory in the Portuguese slope (37.8ºN, 9.4ºW) and travels northward up to the Extremadura Promontory. There, the northward flow converges with the northern closure of the Tagus Anticyclonic Gyre and continues northward forming the poleward slope current that will follow the European continental slope up to Porcupine Bank.

The circulation in Horseshoe Basin is mainly characterized by two opposite zonal currents. The northern half of the basin hosts the westward current originated in Gorringe Bank, confirming the results suggested by Iorga and Lozier (1999a); this flow turns cyclonically once it reaches the western margin of the basin. In the southern half of the basin, the turning flow is channelized eastward under the influence of the mountain chain of Ampere Bank. According to our results this flow continues eastward exiting Horseshoe Basin and returns back to the Gulf of Cadiz. Where it provides part of the water masses that will be entrained by the Mediterranean Water Masses cascading in the Gulf of Cadiz.

The analysis of the mean vorticity field at 1000 m (Figure 3) provides information about the main gyres occurring in the area. It is worth mentioning that this field is affected by the shear of the flow at the vicinity of the continental slopes. This explains the noisy values detected near the continental margins. The strong shear between the descending MOW flow and the surrounding water masses in the Northern slopes of the Gulf of Cadiz generates in this area the highest cyclonic vorticity of the figure. This vorticity values denote the intense mixing processes taking place in the area. The bathymetry also has an important influence: to all the sea mountains included in the domain are associated high negative values of vorticity indicating anticyclonic circulation around them, even when the top of the obstacle is hundreds of meters below the level of 1000 m depth, as it is the case of the sea mount at 36.4ºN and 13.0ºW, whose summit is at 1893m depth. Tagus Basin shows negative values of vorticity mainly related with the presence of the Tagus Anticyclonic Gyre. However, results in Horseshoe basin show the presence of a generalized cyclonic circulation with two separated centres of vorticity located at 14ºW and 12ºW.

Regarding the cyclonic gyre near the Gulf of Cadiz suggested by previous studies (Iorga and Lozier, 1999a, and 1999b), the reanalysis does not provide a clear signal of positive vorticity centred around 35ºN and 9ºW. Results in this area show mainly a zonal westward flow advecting water into the Gulf of Cadiz. In this area, the water transported eastward diverges taking two directions: part of the flow turns northward to be reincorporated into the main MOW current in the Iberian slopes, and a second part of the flow turns anticyclonically following the African slope towards the Seine Abissal Plain. The centre of this circulation pattern appears on the map of vorticity near the slope as an area of negative values at 34.2ºN and 9.2ºW. The splitting of the eastward flow entering the Gulf of Cadiz in two branches at 9.4ºW is favoured by the bathymetric zonal elevation at 35.3ºN in the Gulf of Cadiz.

Volume transports have been computed across the limits of the two boxes defined around Gorringe Bank and Ampere Bank (GBB and ABB respectively). The analysis of transports through the limits of these boxes provide information about the water masses entering/leaving Horseshoe Basin as well as the meridional transport thereof.

Figure 4 shows the mean transverse velocities and net volume transport in the sections defined in GBB and ABB. Results show the main input of MOW into Horseshoe Basin comes through the northern and eastern limits of GBB. The presence of Gorringe Bank highly influences the MOW flow coming from Cape St. Vincent. Part of this flow is deflected towards Horseshoe Basin following the southern flank of Gorringe Bank and enters the basin through the eastern-GBB section (0.7±0.3 Sv). The input through the northern limit of GBB is induced by the seamounts at 36.2ºN and 14.5ºW which forces a southward transport of water from the Tagus Anticyclonic Gyre towards the Horseshoe Basin. The net transport across sections evidences that there is an appreciable flow entering the basin through the eastern-GBB boundary (0.7±0.3 Sv), however the flow through the northern-GBB boundaryt (1.3±0.3 Sv) is almost twice higher and is the main source of MOW in Horseshoe Basin.

The combined transport of water into Horseshoe Basin (northern-GBB + eastern-GBB boundaries) is coherent with the results found in Carracedo et al. (2104). However, while in this work the transports have been computed between two fixed levels, in Carracedo et al. (2014) they are computed for separated water masses. Therefore, the results obtained here are similar to the transport that Carracedo et al. (2014) labelled as Mediterranean Water and recirculated Central Water.

Regarding to the water outputs in GBB, figure 4 shows the presence of a westward transport that overpasses the seamounts and leaves the box by crossing the western boundary. This transport of 0.8±0.3 Sv is the starting of the MOW westward pathway described in bibliography (Iorga and Lozier, 1999a and 1999b). The latitudes where this westward flow occurs range from 35.6ºN to 37.2ºN.

The net meridional transport between the GBB and ABB is shown in Figure 4. The southward transport of water (1.0±0.3 Sv) in Horseshoe Basin is mainly induced by the cyclonic turning of the westward current previously described in the basin, in agreement with the results obtained by Iorga and Lozier (1999a, and 199b). The southward penetration of this water is stopped by the southern wall of the basin redirecting this flow eastward. However, results suggest a net external water input (0.8±0.5 Sv) entering the basin through the southern boundary of ABB. The two flows then converge and exit ABB eastwards towards the Gulf of Cadiz (1.8±0.4 Sv). The θ/S diagrams averaged in GBB and ABB (Figure 4b) show the differences in temperature and salinity between the northern and southern half of Horseshoe Basin. Such differences mainly affect the upper-intermediate layers of ABB where a freshening and cooling of waters is appreciable. The water flow averaged in GBB presents the usual θ/S profile of MOW with a peak in salinity around 1000 m depth (Leadbetter, 2007). On the contrary, the θ/S profile averaged in ABB shows some mixing of AAIW and North Atlantic Deep Water (NADW). The salinity peak is still found at approximately 1000 m depth, however the influence of AAIW is evidenced by the temperature and salinity reduction in the layers above the salinity peak (from 600 to 1000m depth) producing a concavity that sharpens the peak. The influence of NADW is reflected in the freshening and cooling in deeper layers.

Despite of the net southward transport between GBB and ABB boxes (Figure 4a), the averaged velocities across their shared boundary reveal smaller northward flows advecting water from ABB to GBB; this implies the existence of meridional mixing processes in Horseshoe Basin. The northward transport of modified water can be appreciated in the velocity and vorticity fields shown in Figures 2 and 3: the general circulation in Horseshoe Basin is composed by two separated centres of cyclonic vorticity, located at 35.7ºN and 14.2ºW and at 35.7ºN and 12.0ºW. The circulation around these two centres explains the

north-south flows in Horseshoes Basin. The θ/S profile averaged at the Eastern boundary of ABB (Figure 4b) evidences that the properties of water exiting the basin towards the Gulf of Cadiz are the result of the mixing between the southern and northern waters in Horseshoe Basin. However the modification of properties mainly affects the upper-intermediate layers above 1100m depth.

## 4.2 Cape Ghir

The average circulation pattern at 1000 m shown in Figure 2 reveals the existence of a zonal current at 31.5ºN. It is formed along the African continental slope and confirms the existence of an advective westward flow that penetrates into the North Atlantic up to 16.2ºW. This current is associated with an area of cyclonic vorticity west of Cape Ghir and centred at 30.5ºN and 11.6ºW (Figure 3). The domain CGB, near the Cape Ghir, has been defined to analyse the source of this flow (Figure 1). Figure 5 displays the velocity and water transport at the boundaries of CGB. It reveals the entrance of water masses from the north as a weak southward transport (0.6±0.3 Sv) between 12ºW and 11ºW, and from the south as a more intense transport (2.9±0.2 Sv) close to the continental slope. The exit of water masses occurs across the western limit of CGB where a westward current of 5-8 cm/s (3.3±0.3 Sv) pushes the converging water masses towards the outer ocean (Figure 5a). The θ/S diagram averaged in the boundaries of the domain shows the presence of the salinity peak at 1000 m associated to the MOW (Figure 4b). However its temperature and salinity values are significantly lower than ones obtained 500 km to the north at Horseshoe, where the MOW salinity peak ranges between 35.9-36.0 PSU and 10.1-10.4 ºC. The water layers above the salinity peak reveal the strong influence of the AAIW through the presence of a concavity in the profile associated to the cooler and fresher AAIW. The result found in the Northern section of CGB is consistent with the observed climatological θ/S diagram averaged from 10º to 12ºW and from 31º to 33ºN by Machín and Pelegrí (2009). Comparing the θ/S profiles of the Northern and Southern boundaries of the GCB, results show a different relative contribution of MOW and AAIW in each section. While in the Southern-GCB boundary the θ/S profile shows a dominance of AAIW with reduced values of salinity and a concavity in the upper-intermediate waters, at the Northern-GCB section the θ/S diagram shows a greater influence of MOW with a sharper peak of salinity at 1000 m. According to the θ/S diagram averaged at the Western boundary of the box, the outgoing waters reveal a profile influenced by both the water masses of the Northern and Southern boundaries of GCB. The peak of salinity diminishes and the values of temperature and salinity in the upper-intermediate layers increase reducing its concavity. The analysis of the mean velocities shown in Figure 2 provides more information about the circulation processes taking place in this area. South of Cape Ghir the poleward along-slope current mainly composed by AAIW follows the continental slope up to the promontory near Cape Ghir and even further (Machín and Pelegrí, 2009). There, the promontory and the opposition of the southward flow of MOW forces the cyclonic turning of AAIW that detaches from the continental slope and starts an advective westward flow penetrating up to 16.2ºW. The mixing resulting from this confrontation of water masses leads to the modification of temperature and salinity seen in the θ/S profile averaged in Western-GCB boundary.

## 5 Interannual variability

To analyse the interannual variability of MOW in Horseshoe Basin, the anomalies of salinity at 1000 m have been computed and spatially-averaged in the area composed by GBB and ABB. The analysis of the time series of salinity anomalies reveals a different behaviour of the maximum and minimum anomalies (Figure 6). While the maximum values of salinity are spread over the time, the minimum anomalies are concentrated around a unique event occurring during the period 2000-2003. This low salinity event in the area is clear since all values under the 10th percentile of the time series are found within these years. This event is consistent with the results found by Bozec et al. (2011), who report a slight retreat of the inner salinity contours towards the east, especially after 2000.

As stated in section 2, previous works on the variability of hydrographic properties in the MOW reservoir have concluded that changes in MOW properties are not dominated by changes in MSW properties (Lozier and Sindlinger, 2009; and Bozec et al., 2011). This finding is confirmed by the results of this work, in figure 6a which shows salinity anomalies averaged in Horseshoe Basin and CVS. The correlation between both time series is not significative, thus denying direct-linear relationship between salinity anomalies in these areas. This correlation remains barely significative even if the time series are lagged: the maximum significative cross-correlation (0.22) is obtained when the time series averaged in CVS is retarded 2.5 years (Figure 7).

The dates of the minimum and maximum anomalies of salinity defined respectively as the values under/over the 10th/90th percentiles have been used to derive the averaged fields on these dates. The resulting composites, associated to high/low salinity situation have been used to analyse: (1) the θ/S diagram averaged in the Horseshoe Basin (Figure 6b), (2) the meridional section of salinity at 36ºN coinciding with the limit between GBB and ABB (Figure 8), and (3) the salinity distribution at the maximum salinity depth (Figure 9). Since both composites are derived from the fields averaged at dates where the anomaly at 1000m depth is remarkably high or low, they represent the ocean state associated to these particular salinity conditions. Moreover, as every minimum value selected is comprised in the period 2000-2003, the derived minimum composite reflects mainly the ocean state under this specific low salinity event.

The θ/S composites shown in figure 6 reveal that the variability in the Horseshoe Basin affects mainly the layers below the salinity peak. Under conditions of the minimum salinity event, the water masses below 1000 m depth suffer a decrease of approximately 0.25 PSU. Temperature is also affected during this event, showing a cooling of about ~0.6ºC. The structure of the water column is also modified, the salinity peak associated to the MOW core being pushed from the usual 1000 m up to 800 m depth.

Based on three repeated sections at 36ºN, Leadbeter et al. (2007) observed interannual variability of intermediate water masses between 10ºW and 20ºW. This variability mainly affected the layers above the salinity peak, whereas variability in lower levels were smaller. However, our results, based on the use of a high resolution regional reanalysis, suggest an inconsistency since the θ/S variability in this area mainly affects the layers below the MOW salinity peak. As presented in previous sections, variability above the salinity peak can be rather attributed to latitudinal variations, as the differences between θ/S diagrams averaged at GBB and in ABB (~100 Km. southward) are mainly found in the upper levels (above the MOW salinity peak). On

the contrary, according to the present analysis, the main temporal variability in the Horseshoe Basin is found in layers below the MOW tongue. The composite sections of salinity at 36ºN shown in Figure 8 depict the vertical structure of the water column under conditions of maximum and minimum salinity anomaly. Under conditions of the minimum salinity event (Figure 8a), results suggest a westward propagation of the underlying NADW that leads to a retreat of the MOW tongue together with

an upward displacement of the water mass core. This process also implies a general freshening and cooling of the whole water column. The accumulation of low-salinity waters west of 15ºW and below 1100m supports this hypothesis and suggests that penetration of the NADW into Horseshoe basin is limited by the Azores-Portugal Rise. The perturbation in isohalines can be appreciated up to 400 m above the top of the seamount. This result agrees with Bozec et al. (2011) who reported a similar behaviour of the boundary between MOW and the underlying LSW in the central Atlantic.

Figure 9 analyses the maximum of salinity between 500 and 1500 m depth for the composites of maximum and minimum salinity at Horseshoe Basin. As seen previously, during the high salinity situations, the maximum of salinity appears deeper than during the low salinity situations. Additionally the overall salinity field becomes saltier, inducing a westward displacement of isohalines.  The bigger salinity differences are found at the northwest limit of the Horseshoe Basin, near the Azores-Portugal Rise. On the contrary, in the vicinity of the continental platform the salinity differences are smaller, or even negative in the

Gulf of Cadiz and surroundings of Cape St. Vincent. This result suggests a negatively correlated behaviour between the salinity fields of Cape St. Vincent and Gulf of Cadiz, and Horseshoe Basin.

**6 Conclusions**

In the present work, an analysis of the spreading processes of MOW in the North-East Atlantic has been performed through the use of a high-resolution ocean reanalysis: the CMEMS IBI regional reanalysis. The ocean properties and flows are analysed

at intermediate depths (500-1500 m depth) in the Tagus, Horseshoe and Seine basins. These basins are adjacent to the Gulf of Cadiz and they form the main area where the MOW accumulates before spreading into the North Atlantic. In this work we analyse the tongue of MOW in the reservoir area, the influence of bathymetry over the spreading of MOW, and its interactions with the surrounding water masses (NADW and AAIW). This work has also been conducted, comparing the IBI reanalysis with previous works on the characterization of MOW in the North Atlantic. The high agreement of results with the known

features of MOW suggests a proper reproduction of the dynamic features of the intermediate waters in the region. Moreover, the high resolution of the IBI reanalysis product allows to describe new mesoscale features, not previously reported in literature.

One of the main contributions of this work results from the updated description of circulation patterns of MOW in the east North Atlantic. A coarse description of the circulation patterns in this area was reported by previous studies (Daniault et al.,

1994; Mazé et al, 1997; Iorga and Lozier, 1999a; 1999b; van Aken, 2000), however the high resolution of IBI reanalysis allows a more detailed description of the circulation in the region. Figure 10 summarizes the circulation patterns described by the abovementioned works (black arrows) and the updated scheme, resulting from the IBI reanalysis data (represented by red

arrows). Once the recently formed MOW overpasses Cape St. Vincent, the presence of Gorringe Bank splits the flow in two branches: one of them enters Tagus Basin, describing an anticyclonic gyre, whereas the other branch of MOW flows westward along the northern boundary of Horseshoe Basin. The bathymetry also forces the cyclonic circulation in Horseshoe Basin, the westward flow iniciated in Gorringe Bank turns cyclonically when it encounters the seamounts at the western boundary of the basin (the Azores-Portugal Rise). Thereafter, water masses recirculate eastwards towards the Gulf of Cadiz. Seamounts of the Ampere mountain chain lead to the formation of anticyclonic vorticity in the proximity of these obstacles, and circulation around these centres favours active transports of southern water into Horseshoes Basin. The external water entrained is mainly composed by a diluted form of AAIW in the upper-intermediate layers and NADW at depths below 1000 m. Thereby, the MOW water masses that recirculates into the Gulf of Cadiz are previously modified in the Horseshoe region through mixing with AAIW and NADW. This process could explain the presence of an enhanced concentration of nutrients in the MOW, as reported by Van Aken (2000).

The interaction between MOW and the deeper AAIW near the African continental slope is highly influenced by the bathymetric promontory between Cape Ghir and Cape Sim. The converging flows of MOW, which travel southward along the African Continental slope, and of AAIW that enters northwards the basin through a narrow gateway between Fuerteventura and the African Shelf, collide in this area producing a zonal transport of mixed MOW-AAIW waters towards the inner ocean. The potential role of this branch as an advective pathway for MOW into the subtropical gyre and its relationships with the Madeira Eddy Corridor, described by Sangrá et al. (2009), will be analysed in future works.

The analysis of the circulation patterns in the region has highlighted the role of the bathymetry as a key factor determining the spreading and mixing patterns of MOW in the region. The presence of three abyssal plains separated by seamount chains implies an orographic complexity that highly influences the processes in the area. This work has also reported some cases where the bathymetric features can modify the flow hundreds of meters above the obstacle, some topographic features moreover coinciding with the observed areas of meddy formation as described by previous authors (Sangrá et al, 2009) and suggesting some influence of bathymetric obstacles on the meddy formation. Therefore, in other to obtain a realistic representation of the ocean, the influence of the high resolution bathymetry must be considered by modelling studies in this region and depths.

The analysis of the CMEMS IBI currents and transports together with the derived hydrographic patterns has allowed the reconstruction of the dynamic variability in Horseshoe Basin. The composite analysis of temperature and salinity in the region leads to the conclusion that the main source of interannual variability in the Horseshoe comes from the deeper layers of MOW. Our work has shown that the boundary between the MOW and the underlying NADW is subject to interannual variability. As far as observed within the limited temporal extension of the CMEMS IBI reanalysis (25 years), the presence of the MOW tongue in Horseshoe Basin seems to be the normal situation. However, the reanalysis reveals the existence of a remarkable event (2000-2004) where the NADW advances into Horseshoe Basin. Under these specific conditions, the MOW core retreats eastwards, diminishing the salinity in the whole water column of Horseshoe Basin. The analysis of the maximum vertical salinity of composites as led to conclude that the North Atlantic circulation influences the water properties of the Gulf of Cadiz

and Cape St. Vincent. Thus, the advance of NADW imply the accumulation of MOW water near its source. Therefore, the low salinity event in Horseshoe Basin is associated to an increase of salinity in the intermediate layers of the Gulf of Cadiz and Cape St. Vincent.

**Acknowledgements.** The authors thank to Copernicus Marine Environment Monitoring Service for providing the data. Additionally, the helpful comments of Karen Guihou and the other referees are gratefully acknowledged.

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

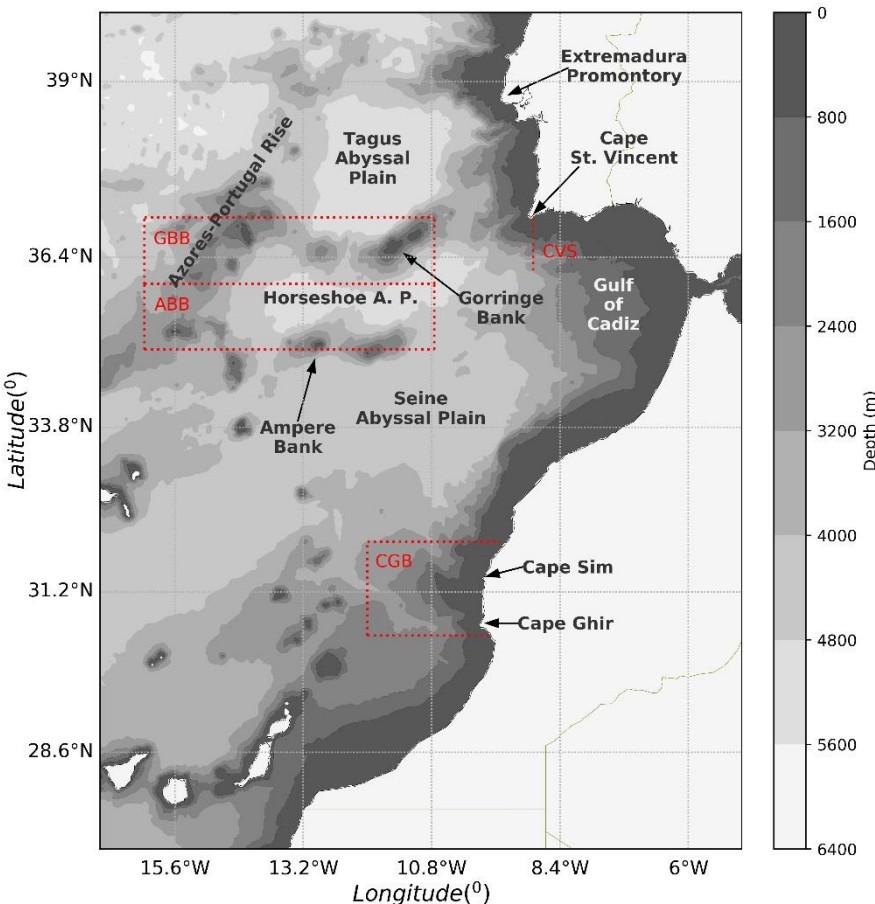

**Figure 1: Map of the study domain showing the geographical features mentioned in the text. Red dotted lines denote the sections where the transports have been computed. They define the three study boxes used in the present work: Gorringe Bank Box (GBB), Ampere Bank Box (ABB), and Cape Ghir Box (CGB) as well as a meridional section in the continental platform: Cape St. Vincent Section (CVS).**

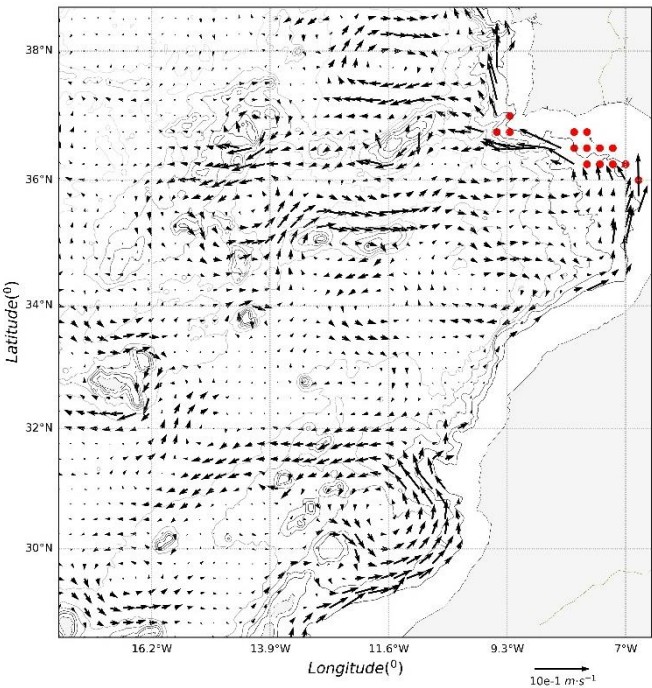

**Figure 2: Velocity field at 1000m given by the CMEMS IBI reanalysis. For clarity reasons, grid points with very high relative velocities have been masked near the southern Iberian platform.**

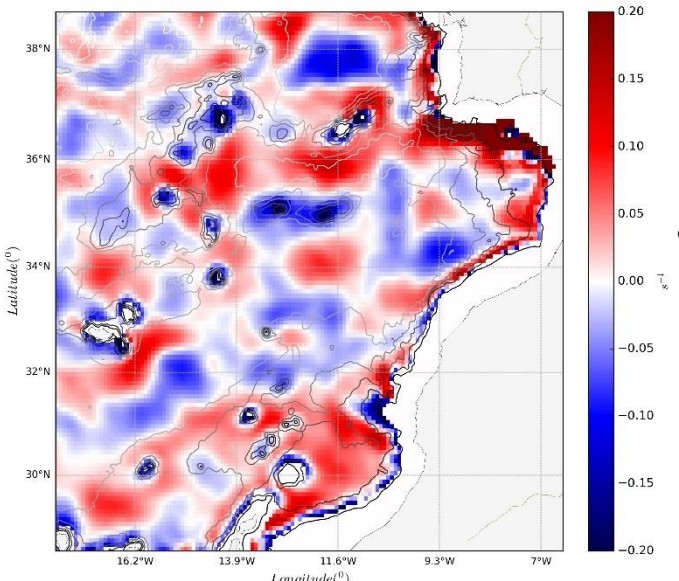

**Figure 3: Vorticity field at 1000m derived from IBI velocities. Red/blue colours denote cyclonic/anticyclonic vorticity, respectively.**

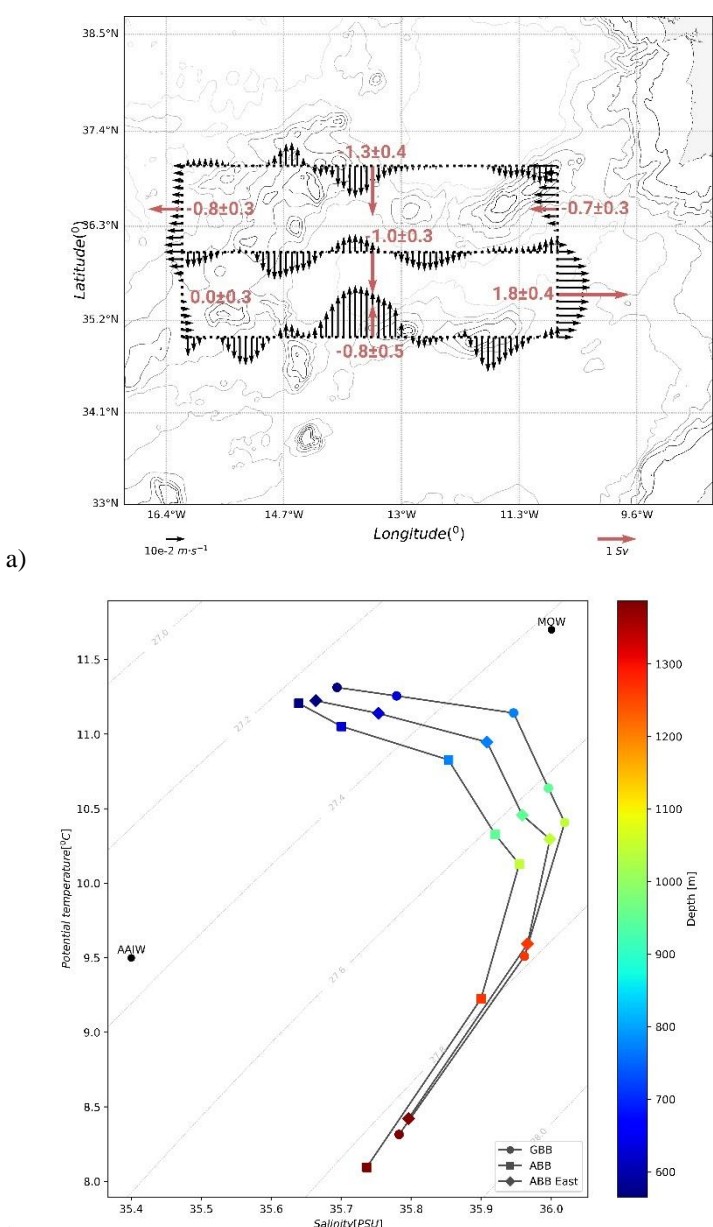

a)

b)

**Figure 4: (a)** Transverse velocity (black arrows) and net volume transport (red arrows) in sections delimiting GBB and ABB. **(b)** Mean θ/S diagram averaged in GBB (circles), ABB (squares) and along the section marking the Eastern-ABB boundary (diamonds). Dotted isolines correspond to potential density anomaly in Kg/m$^3$. Black dots represent the position of the source water types AAIW and MOW.

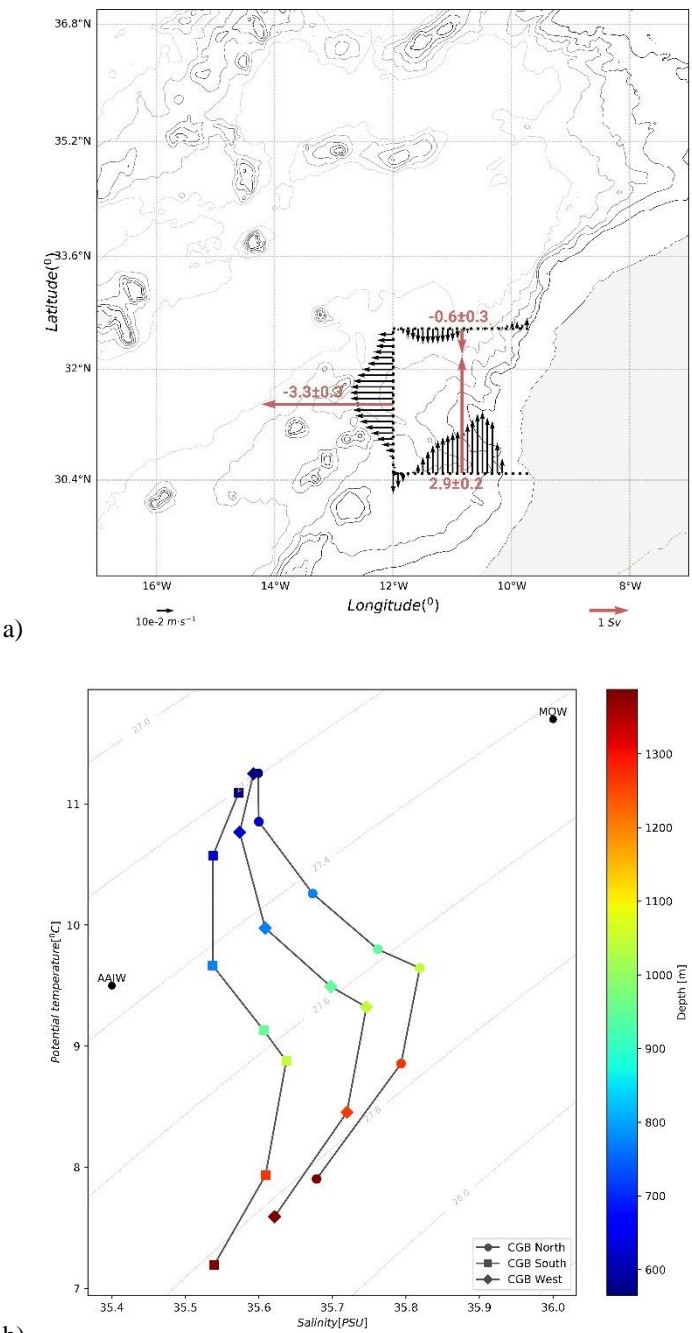

a)

b)

**Figure 5: (a) Transverse velocity (black arrows) and net volume transport (red arrows) in sections delimiting CGB. (c) Mean θ/S diagram averaged along the sections delimiting the Northern-CGB boundary (circles), Southern-CGB boundary (squares) and Western-CGB boundary (diamonds). Dotted isolines correspond to potential density anomaly in Kg/m³. Black dots represent the position of the source water types AAIW, and MOW.**

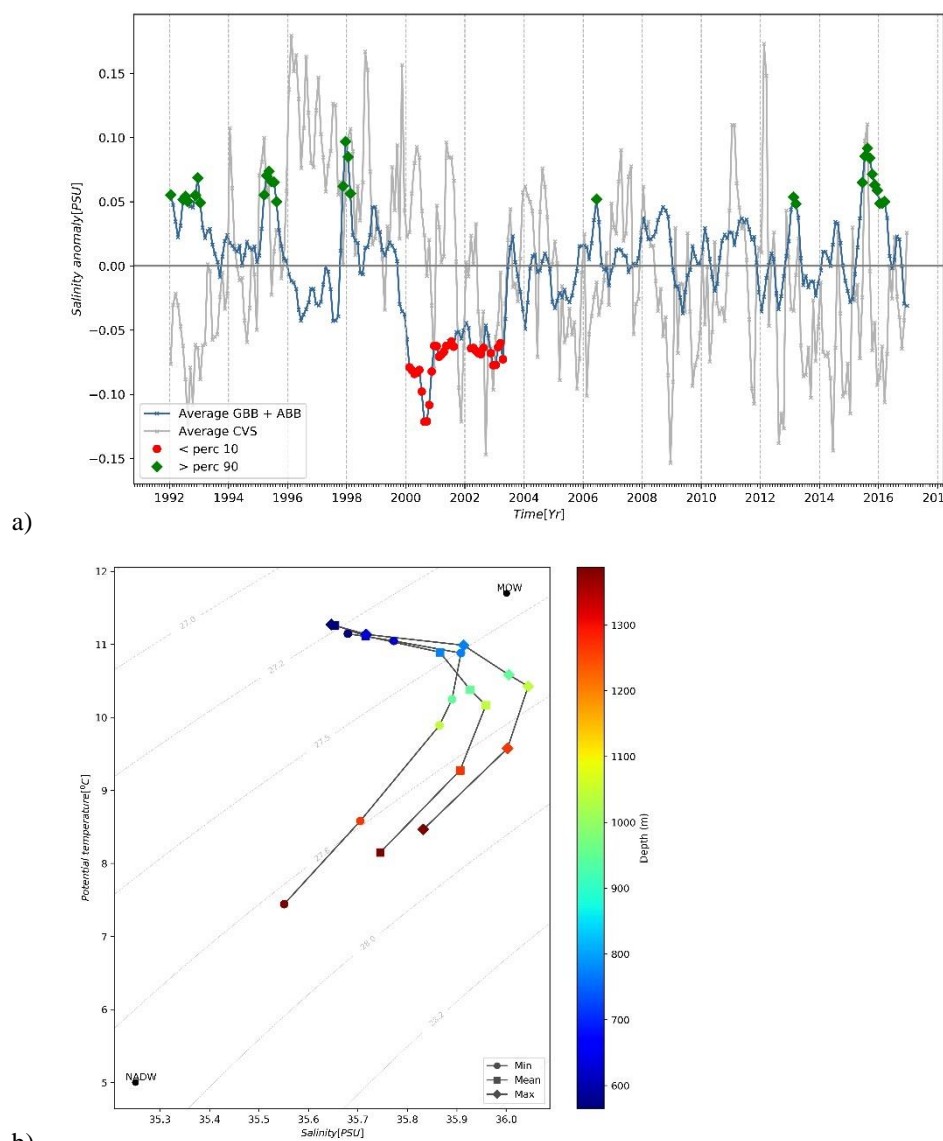

a)

b)

**Figure 6: (a) Anomalies of salinity at 1000 m averaged in the whole Horseshoe Basin (areas GBB and ABB combined) and in CVS (blue and grey lines, respectively). Red/green dots depicts the values under/over the 10th/90th percentile of salinity anomaly in Horseshoe Basin. (b) θ/S diagram in Horseshoe Basin (areas GBB and ABB combined) averaged in the complete time record (squares), dates of minimum salinity (circles), and dates of maximum salinity (diamonds). Dotted isolines correspond to potential density anomaly in Kg/m³. Black dots represent the position of the source water types NADW and MOW.**

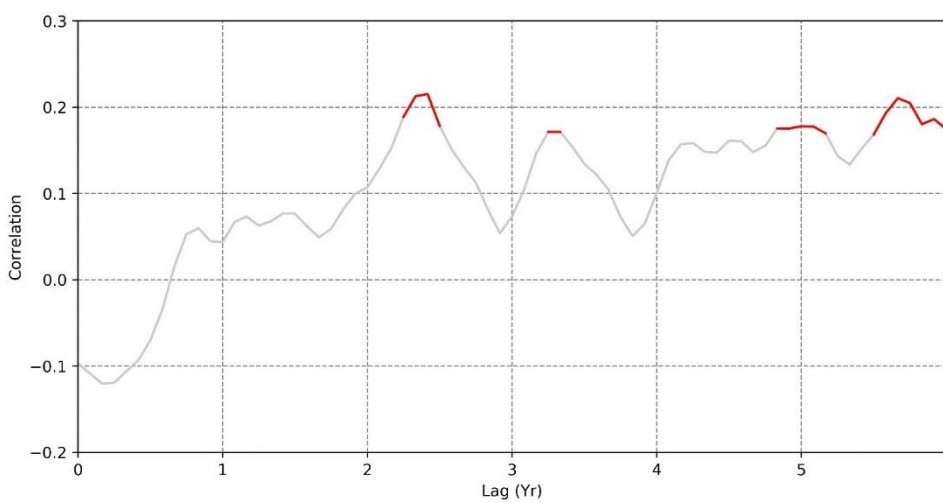

**Figure 7: Cross-correlation coefficient of salinity anomalies averaged in Horseshoe Basin (areas GBB and ABB combined) and CVS. Lag corresponds to the displacement of the CVS time series respect to the Horseshoe Basin time series. Red segments denote statistically significant correlations with significance level of 99%.**

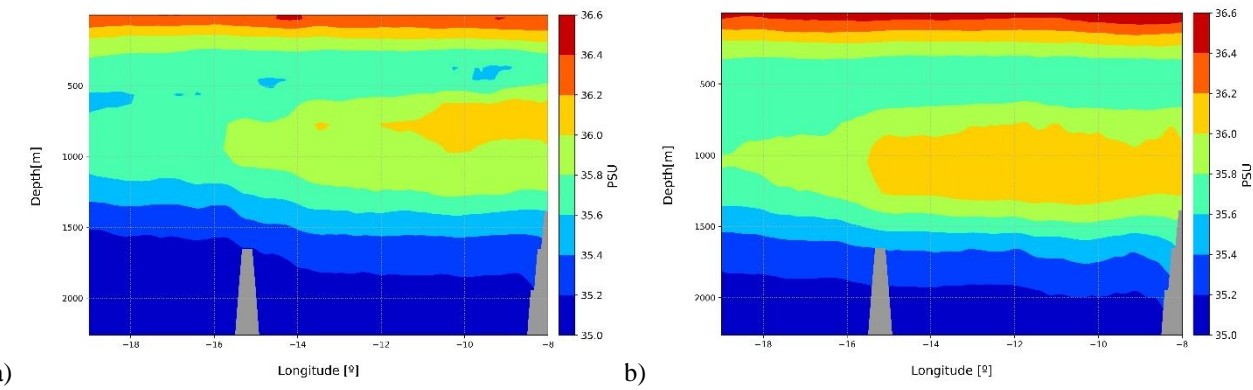

a)                                            b)

**Figure 8: Section of salinity at 36ºN (coinciding with the limit between GBB and ABB) of the composites resulting from the averaged fields defined from the 10th (a) and 90th (b) percentiles of salinity anomalies in Horseshoes Basin.**

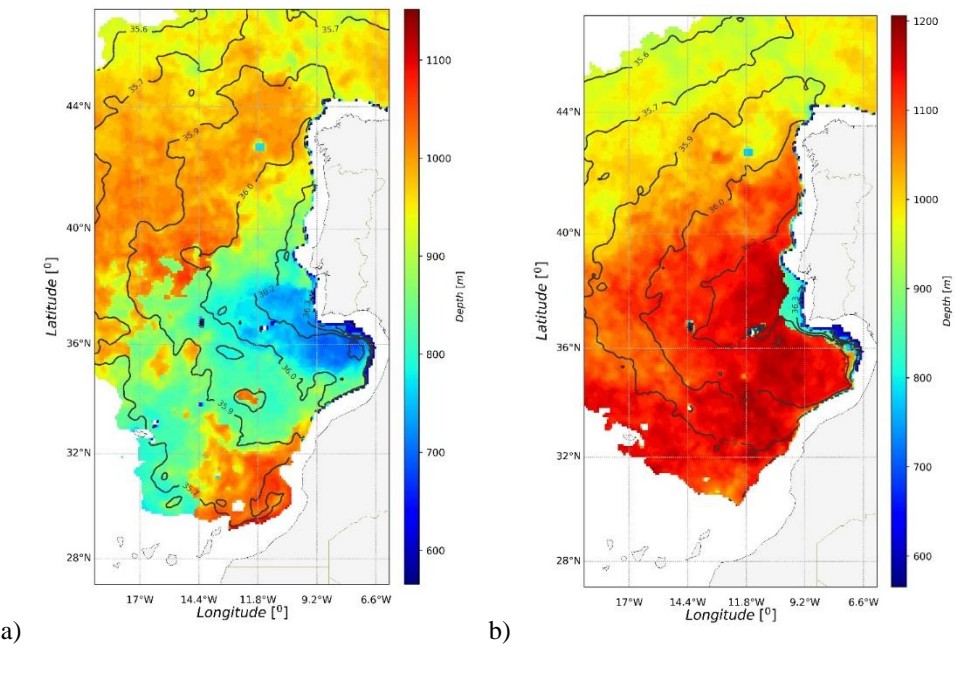

a)                                                          b)

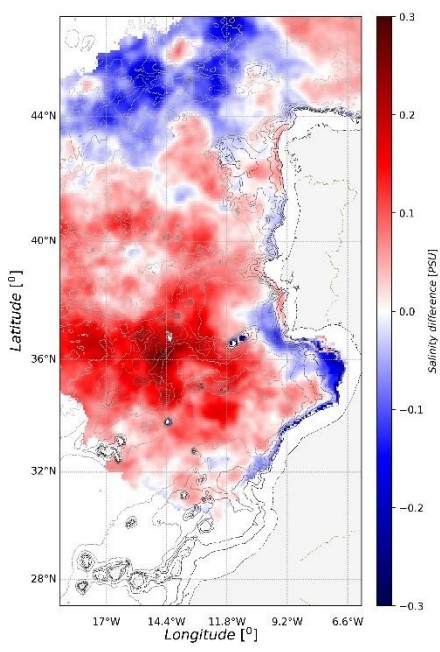

c)

**Figure 9: (a and b) Maximum salinity depth (shaded colours) between 500 and 1500 m and salinity (isolines) at the maximum salinity depth. Maps obtained from the composites of minimum (a) and maximum (b) salinity defined by 10[th] and 90[th] percentiles of salinity anomalies in Horseshoes Basin respectively. (c) Difference of salinity between panels a and b, contour lines represent bathymetry.**

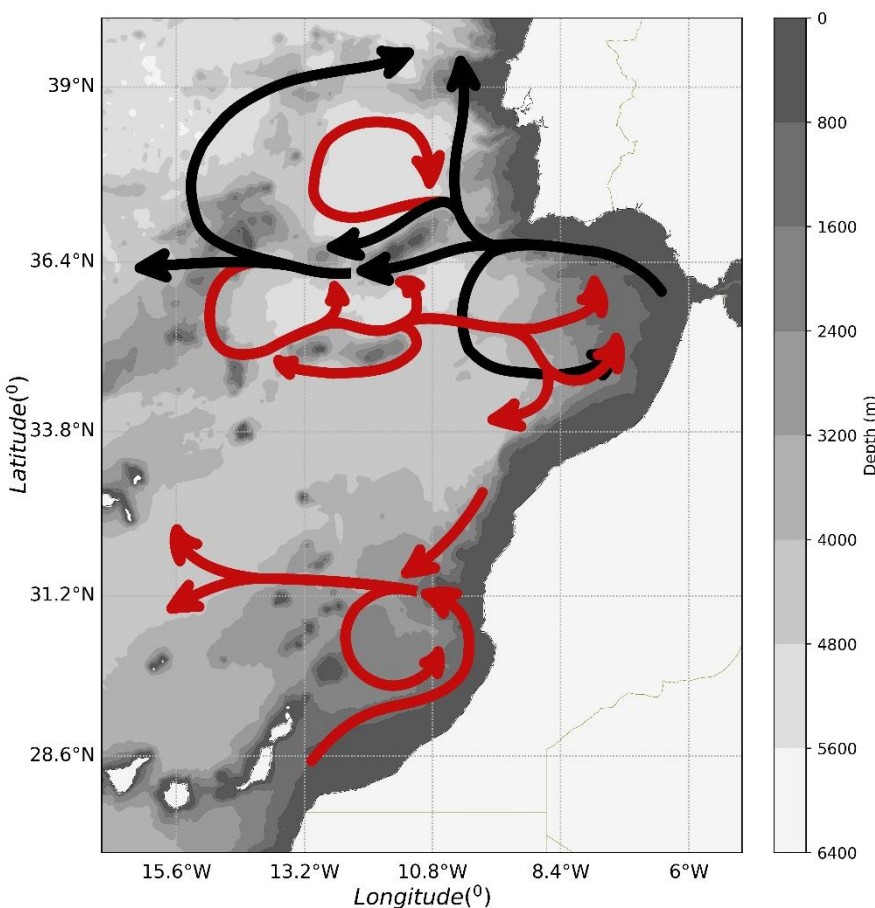

**Figure 10: Schematic representation of the overall Mediterranean Outflow Water pathways in the eastern North Atlantic. Black arrows show the current scientific consensus according to Iorga and Lozier (1999a) and Carracedo et al. (2014). Red arrows summarize other MOW features described in the work. Results based based on an analysis of the CMEMS IBI reanalysis (25 years record and 1/12º resolution).**