# Peer review of "Regional circulation patterns of Mediterranean Outflow Water near the Iberian and African continental slopes"

_Ocean Science, 2018_

## Referee Comment (RC1) · Anonymous Referee #1 · 1 Mar 2019

GENERAL COMMENTS:

The manuscript "Regional circulation patterns of Mediterranean Outflow Water near the Iberian and African continental slopes" by Pascual-Collar et al. shows an application of CMEMS-IBI reanalysis to study the behaviour of Mediterranean Outflow Water (MOW) in that region. It is an example of the strong potential being developed at CMEMS for studying climate-ocean processes with a regional high spatial resolution taking into account mesoscale features.

The topic is fully within OS fields of interest, and authors present an application of a novel and powerful tool, CMEMS high-resolution ocean reanalysis. The manuscript is

well structured, and it has a sound methodology and provides interesting results. As scientific novelties authors report the ability of resolving mesoscale features of MOW circulation, allowing to study the influence of topography in MOW spreading and the MOW interaction with surrounding water masses. However, there is a way too large number of typos, spelling and grammatical mistakes, which do require a careful proof-reading and correction. Also, I have some specific comments/suggestions, which I think can improve the overall good quality of the manuscript after a minor/moderate revision.

SPECIFIC COMMENTS:

INTRODUCTION

1. The state of the art is adequately described in the Introduction. However, there are at least two recent works that should be included. These are the works of Carracedo et al. (2014) and Izquierdo and Mikolajewicz (2019) (the latter is just published, so it is clear that authors could not be aware of it).

Carracedo et al. (2014) described the circulation in the Azores-Gibraltar strait-Canary region by means of climatological data. Their box analysis shows the relative importance of the MOW northward and westward advective branches and their seasonality, as well as the interplay between the different water masses present in the region (including mixing). Izquierdo and Mikolajewicz (2019) present a process study based on a high-resolution (similar to the presented here) model output. They also show mesoscale features of MOW circulation in the region and stress the importance of topography (through tide-topography interaction) in MOW spreading. These all are very relevant topics for this study and taking them into account in the discussion of the obtained results will improve the manuscript.

SECTION 4:

2. How are volume transports computed? I have not found indication about it, nor

about the upper and lower limits for the vertical integration. In the Conclusion (page 11, line 22) there is a mention of 500-2000 m depth, but clearly it must be explicitly indicated before.

3. Please, provide confidence intervals for the calculated mean quantities.

SECTION 5:

4. The interannual variability in the Horseshoe Basin is discussed without taking into consideration the possible variability of MW outflow rate and properties at the Strait of Gibraltar. A plot showing the temporal evolution of the MW outflow at the Strait, and the salinity anomalies would provide arguments to neglect this potentially relevant factor.

5. I would add a figure showing the salinity distribution at the maximum salinity depth for the mean state and the positive and negative anomalies (for example, salinity color-coded and maximum salinity depth with overlaid contours).

SECTION 6:

6. In general, I personally prefer a separate section dealing with the discussion of the results. However, I understand that other options are possible. In any case here or in a corresponding previous section, I suggest authors to discuss the circulation features, the net volume transports and mixing processes details taking into account the results from Carracedo et al. (2014) and Izquierdo and Mikolajewicz (2019) where possible.

TECHNICAL CORRECTIONS:

The list is not exhaustive. Please, check spelling and grammar carefully.

P1L9: Strait of Gibraltar instead of Gibraltar Straight P1L11: remove "depth" P1L11: plays a key role instead of is a key role P1L13: Remove "of this" P1L13: do instead of does P1L19: reveal i.o. reveals P2L3: Danieault i.o. Denieault (and elsewhere). P2L28: Iorga i.o. iorga P3L18. reveal i.o. reveals P3L28: shelf i.o. self P3L31: "exceeding the 500 m depth" is wrong sense. Reword it P4L7: 1999b i.o 199b P7L5-

7: Use . i.o , P7L8: Please, homogenise the writing of Cape St-Vincent (see line 19) P7L21 mass i.o. masses (or change the verb third person) P9L8: Figure 4b i.o. Figure 3b P9L20: south i.o. South P10L16: This statement is redundant, tautological. P10L32: delete 0.75. I do not appreciate such difference in Fig. 6

References: Carracedo, L., Gilcoto, M., Mercier, H. Pérez F. (2014) Seasonal dynamics in the Azores-Gibraltar Strait region: A climatologically-based study. Progress in Oceanography 122, 116-130.

Izquierdo, A. , Mikolajewicz U. (2019) The role of tides In the spreading of Mediterranean Outflow Waters along the Southwestern Iberian Margin, Ocean Modelling 133, 27-43.

---

## Referee Comment (RC2) · Anonymous Referee #2 · 7 Mar 2019

Review of the paper entitle:

"Regional circulation patterns of Mediterranean Outflow Water near the Iberian and African continental slopes" by Álvaro de Pascual-Collar, Marcos García-Sotillo, Bruno Levier, Roland Aznar, Pablo Lorente, Arancha Amo, Enrique Fanjul.

The manuscript tackle an important and relevant scientific issue dealing with the analysis of the spreading processes of MOW in the East North Atlantic using a high resolution (1/12°, about 8 Km) CMEMS IBI ocean regional reanalysis. Focusing their analysis for the intermediate layers (500-2000m depth) in the Tagus, Horseshoe and Seine basins respectively.

[Figure]

This study is very interesting because increase understanding on several circulation features that reveals the important role of bathymetry in the spreading of MOW when its leave the Gulf of Cadiz. Moreover, the temporal coverage of the IBI reanalysis (1992-2016) permitted the analysis of the temporal variability resulting in the description of the mid-term variability of MOW in the region.

The manuscript is also relevant for ocean climate variability studies and in particular for research and simulation of the interaction between Mediterranean Sea and North Atlantic ocean that have to be considered as a unique ocean/climate system (Artale et al., 2006).

For all these reasons that the results of this paper are very interesting for the oceanographic communities and in particular for those scientists more implicated on the Mediterranean-Atlantic interaction. However the present version still have a lot of a weak points and therefore is not ready to be published for the following reason.

General comments and Major revision:

The scientific matter of the manuscript isn't a really new augment; actually in the literature there are many example on this, either in the modelling field or in the analysis of the in situ observations. But the manuscript has many strong/weak points and many novelty elements the most relevant one are the following:

• The manuscript encompassing a comprehensive introduction, but the role of the Gibraltar Strait (and tide) is completely missed (this can explain the seasonal signal (20%) of the overall interannual variability as discussed by the authors) even if this region is resolved and included in the IBI model domain;

• Very interesting the discussion on the concept of (salt) reservoir (tipping point) that resolves many problems for understand the role of the MOW in the Atlantic THC variability;

• Good the use of the CMEMS IBI, but also could be useful other model of CMEMS

for example the Mediterranean Sea Model or using the entire IBI domain that include almost the entire Western Mediterranean Sea;

• Very interesting the updated vision (Figure 8) redistribution of the salt due to the circulation, but unfortunately the dependence of these features from the variability of the source water at Gibraltar Strait is again missed; but is very interesting the role of the bathymetry on this redistribution and specifically the role as salt reservoir of the Tagus Abyssal Plain.

Among the weak points the following one is the most relevant:

• The authors do not exploit the potential of the data available to study the impact of the Gibraltar Strait on the interannual variability of the salt anomaly into the North Atlantic (MOW) due more likely to the non-linear interaction among the hydraulic control modulated by the tide and to the reservoir of MDW downstream of Gibraltar Strait and the Bernoulli suction as a mechanism of transport of this anomaly amount of salt upstream of the Gibraltar Strait.

• Is matter of fact that the inflow/outflow is regulated by the physic of the Strait of Gibraltar and that the properties of the source water of the MW that will be later observed in the North Atlantic still maintaining the memory of the originated Mediterranean water, in fact, following Fig. 4 of Fusco et al, 2008 or Bozec et al., 2011, is very evident the impact on the MOW hydrological value of the quasi-periodical extraction and evacuation of WMDW from the Mediterranean into the Atlantic. Therefore, should be very interesting to verify the hydrological characteristic of MOW in the Gulf of Cadiz and its interannual variability in relation of those observed in Mediterranean Sea Deep Water.

Minor revision:

• There are many typos, please check with the English dictionary;

• In the text at page 9 the figure 3b is 4b;

• At page 9 line 16-23 the sentences aren't supported by the analysis, please clarify;

• In the T/S diagram the value of density curve is completely missed and please put in the T/S mean value that characterize the water types AAIW and NADW, this should be very useful to evaluate at least graphically the mixing of the MOW with these water types/water masses.

References.

1. Fusco, G., Artale V., Cotroneo Y.; Thermohaline variability of Mediterranean Water in the Gulf of Cadiz over the last decades (1948-1999), Deep Sea Research Part I: Oceanographic Research Papers, Volume 55, Issue 12, Pages 1624-1638, 2008; 2. . Artale, V., S. Calmanti, P. Malanotte‐Rizzoli, G. Pisacane, V. Rupolo, and M. Tsimplis (2006), The Atlantic and Mediterranean Sea as connected systems, in Mediterranean Climate Variability, edited by P. Lionello, P. Malanotte‐Rizzoli, and R. Boscolo, pp. 283–322, Elsevier, Oxford, U. K..

Please also note the supplement to this comment: https://www.ocean-sci-discuss.net/os-2018-143/os-2018-143-RC2-supplement.pdf

---

## Author Comment (AC1) · 4 Apr 2019

We gratefully thank your comments and suggestions, we have found some of them very interesting and constructive.

Taking into consideration all the comments received, a new version of the paper has been drafted.

Following we respond to your comments point-by-point:

=====================================================================
=====================================================================

[Figure]

COMMENT:

1. The state of the art is adequately described in the Introduction. However, there are at least two recent works that should be included. These are the works of Carracedo et al. (2014) and Izquierdo and Mikolajewicz (2019) (the latter is just published, so it is clear that authors could not be aware of it).

Carracedo et al. (2014) described the circulation in the Azores-Gibraltar strait-Canary region by means of climatological data. Their box analysis shows the relative importance of the MOW northward and westward advective branches and their seasonality, as well as the interplay between the different water masses present in the region (including mixing). Izquierdo and Mikolajewicz (2019) present a process study based on a high-resolution (similar to the presented here) model output. They also show mesoscale features of MOW circulation in the region and stress the importance of topography (through tide-topography interaction) in MOW spreading. These all are very relevant topics for this study and taking them into account in the discussion of the obtained results will improve the manuscript.

REPLY:

The authors thank the suggestion, we found these studies very interesting and strongly related to the topic. We have introduced several references to this works in the text. Especially to Carracedo et al.

========================================================================
========================================================================

COMMENT:

2. How are volume transports computed? I have not found indication about it, nor about the upper and lower limits for the vertical integration. In the Conclusion (page 11, line 22) there is a mention of 500-2000 m depth, but clearly it must be explicitly indicated before.

REPLY:

We agree with the reviewer, the information of the layer where the transports were calculated was missing, this information has been included in section 3.

The transports are computed from 500 m to 1500 m depth at the lateral faces of each grid cell. The lateral surface of grid cells is computed from the distance between cell corners and the layer thickness.

=======================================================================
=======================================================================
COMMENT:

3. Please, provide confidence intervals for the calculated mean quantities.

REPLY:

We have included the transports and confidence intervals in figures 4a, and 5a. The confidence intervals have been computed by bootstrapping.
=======================================================================
=======================================================================
COMMENT:

4. The interannual variability in the Horseshoe Basin is discussed without taking into consideration the possible variability of MW outflow rate and properties at the Strait of Gibraltar. A plot showing the temporal evolution of the MW outflow at the Strait, and the salinity anomalies would provide arguments to neglect this potentially relevant factor.

REPLY:

Several works have analyzed the variability of properties in the MOW reservoir (Lozier and Sindlinger, 2009; and Bozec et al., 2011), they conclude that the hydrographic properties in the MOW reservoir are not strongly linked with the variability of Mediterranean Water in the Strait of Gibraltar. Actually according to Bozec et al. (2011): "the

observed salinity changes in the MOW reservoir can be explained solely by circulation‚ÄîÇRinduced shifts in the salinity field in the eastern North Atlantic basin"

However, we consider this an interesting discussion point, therefore, following the reviewer suggestion, we have included a new meridional section in the work (See section CVS in figure 1), analyzing the statistical relationship of salinity anomalies between the CVS section and Horseshoe Basin. See figures 6a, and 7.

===================================================================
===================================================================
COMMENT:

5. I would add a figure showing the salinity distribution at the maximum salinity depth for the mean state and the positive and negative anomalies (for example, salinity colorcoded and maximum salinity depth with overlaid contours).

REPLY:

The proposed result has been included (Figure 9) and discussed in section 5. Some interesting results have arisen as a result of this suggestion.

===================================================================
===================================================================
COMMENT:

6. In general, I personally prefer a separate section dealing with the discussion of the results. However, I understand that other options are possible. In any case here or in a corresponding previous section, I suggest authors to discuss the circulation features, the net volume transports and mixing processes details taking into account the results from Carracedo et al. (2014) and Izquierdo and Mikolajewicz (2019) where possible.

REPLY:

While the transports estimated in this work correspond to the net transport between two fixed levels (500-1500), the transports computed in Carracedo et al. correspond to

specific water masses. Moreover, the sections where the transports are computed in each work are often quite distant. Therefore, the direct comparison of results between both works becomes difficult and must be carefully done. However, we have included a paragraph where we compare some features described in Carracedo et al. with the results provided by IBI reanalysis (See section 4.1).

The interest of this work is focused on the spreading of MOW beyond the Gulf of Cadiz. Therefore, despite Izquierdo and Mikolajewicz is very interesting, we have not found many connection points between both works. However, we have included this reference in the introduction.

========================================================================
========================================================================
COMMENT:

The list is not exhaustive. Please, check spelling and grammar carefully.

P1L9: Strait of Gibraltar instead of Gibraltar Straight P1L11: remove "depth" P1L11: plays a key role instead of is a key role P1L13: Remove "of this" P1L13: do instead of does P1L19: reveal i.o. reveals P2L3: Danieault i.o. Denieault (and elsewhere). P2L28: Iorga i.o. iorga P3L18. reveal i.o. reveals P3L28: shelf i.o. self P3L31: "exceeding the 500 m depth" is wrong sense. Reword it P4L7: 1999b i.o 199b P7L5-7: Use . i.o , P7L8: Please, homogenise the writing of Cape St-Vincent (see line 19) P7L21 mass i.o. masses (or change the verb third person) P9L8: Figure 4b i.o. Figure 3b P9L20: south i.o. South P10L16: This statement is redundant, tautological. P10L32: delete 0.75. I do not appreciate such difference in Fig. 6

REPLY:

We sincerely apologize for all these mistakes and typos. We have put an extra effort to improve the English.
========================================================================

Please also note the supplement to this comment:
https://www.ocean-sci-discuss.net/os-2018-143/os-2018-143-AC1-supplement.pdf

---

## Author Comment (AC2) · 4 Apr 2019

We gratefully thank your comments and suggestions, we have found some of them very interesting and constructive.

Taking into consideration all the comments received, a new version of the paper has been drafted.

Following we respond to the comments point-by-point:

=====================================================================
=====================================================================

[Figure]

COMMENT:

General comments and Major revision:

The scientific matter of the manuscript isn't a really new augment; actually in the literature there are many example on this, either in the modelling field or in the analysis of the in situ observations. But the manuscript has many strong/weak points and many novelty elements the most relevant one are the following: The manuscript encompassing a comprehensive introduction, but the role of the Gibraltar Strait (and tide) is completely missed (this can explain the seasonal signal (20%) of the overall interannual variability as discussed by the authors) even if this region is resolved and included in the IBI model domain; Very interesting the discussion on the concept of (salt) reservoir (tipping point) that resolves many problems for understand the role of the MOW in the Atlantic THC variability; Good the use of the CMEMS IBI, but also could be useful other model of CMEMS for example the Mediterranean Sea Model or using the entire IBI domain that include almost the entire Western Mediterranean Sea; Very interesting the updated vision (Figure 8) redistribution of the salt due to the circulation, but unfortunately the dependence of these features from the variability of the source water at Gibraltar Strait is again missed; but is very interesting the role of the bathymetry on this redistribution and specifically the role as salt reservoir of the Tagus Abyssal Plain.

Among the weak points the following one is the most relevant: The authors do not exploit the potential of the data available to study the impact of the Gibraltar Strait on the interannual variability of the salt anomaly into the North Atlantic (MOW) due more likely to the non-linear interaction among the hydraulic control modulated by the tide and to the reservoir of MDW downstream of Gibraltar Strait and the Bernoulli suction as a mechanism of transport of this anomaly amount of salt upstream of the Gibraltar Strait. Is matter of fact that the inflow/outflow is regulated by the physic of the Strait of Gibraltar and that the properties of the source water of the MW that will be later observed in the North Atlantic still maintaining the memory of the originated Mediterranean water, in fact, following Fig. 4 of Fusco et al, 2008 or Bozec et al., 2011, is very

evident the impact on the MOW hydrological value of the quasi-periodical extraction and evacuation of WMDW from the Mediterranean into the Atlantic. Therefore, should be very interesting to verify the hydrological characteristic of MOW in the Gulf of Cadiz and its interannual variability in relation of those observed in Mediterranean Sea Deep Water.

REPLY:

Unfortunately, the in-depth analysis of the influence of hydrological properties of the source water in the Strait of Gibraltar over the MOW reservoir is out of the scope of this work. Additionally, several previous studies stated that the variability of properties in the MOW reservoir is not dominated by the changes in the Gibraltar Strait (Lozier and Sindlinger, 2009; and Bozec et al., 2011). Paraphrasing conclusions in Bozec et al.: "In an observational analysis, Lozier and Sindlinger [2009] showed that the variability of MSW and NACW is too weak to explain the observed MOW variability. [...] Thus, our work has shown that the observed salinity changes in the MOW reservoir can be explained solely by circulation-induced shifts in the salinity field in the eastern North Atlantic basin".

However, we consider this hypothesis an interesting point of discussion. Therefore we have included an analysis of the statistical relationships between salinity anomalies averaged in Horseshoes Basin and south of the Cape St. Vincent (section CVS in figure 1). The results are discussed in section 5. Obviously, the analysis presented does not explore the non-linear dependencies. However, the complexity of analysing non-linear dependencies makes this issue the objective of a future study.

Additionally, part of the results seen in section 5 (figure 9) implies the influence of the North Atlantic circulation over the water properties of the Gulf of Cadiz and Cape St. Vincent.
=====================================================================
=====================================================================
COMMENT:

-There are many typos, please check with the English dictionary.

REPLY:

We apologize for the typos. An extra effort has been put to check spelling and grammar.
========================================================================
========================================================================
COMMENT:

-In the text at page 9 the figure 3b is 4b.

REPLY:

This typo has been corrected, the rest of the figure references has been checked.
========================================================================
========================================================================
COMMENT:

-At page 9 line 16-23 the sentences aren't supported by the analysis, please clarify.

REPLY:

Since the reviewer does not point to any specific issue, we have included some minor changes to clarify the paragraph. However, we find it clear and supported by the results shown in figures 2, 3, and 4. From our point of view:

1.- The distribution of velocities and transport in the shared boundary of boxes GBB and ABB entails the mixing of water masses from south to north and vice versa (Figure 4a).

2.- The T/S profiles shown in figure 4b implies the existence of different water properties (especially affecting the intermediate layers up to 1200m) in GBB and ABB.

3.- The modified water masses exiting the Horseshoe Basin through Eastern-ABB

boundary can be seen in figure 4b.

4.- The existence of two circulation centres in the basin is evidenced in figures 2 and 3.

We would be open to discuss and modify the paragraph in case the reviewer provides some specific arguments that may neglect the results described.
========================================================================
========================================================================
COMMENT:

-In the T/S diagram the value of density curve is completely missed and please put in the T/S mean value that characterize the water types AAIW and NADW, this should be very useful to evaluate at least graphically the mixing of the MOW with these water types/water masses.

REPLY:

We thank the suggestion, it has been applied in figures 4b, 5b, and 6b.
========================================================================
========================================================================
COMMENT:

References. 1. Fusco, G., Artale V., Cotroneo Y.; Thermohaline variability of Mediterranean Water in the Gulf of Cadiz over the last decades (1948-1999), Deep Sea Research Part I: Oceanographic Research Papers, Volume 55, Issue 12, Pages 1624-1638, 2008;

2. Artale, V., S. Calmanti, P. Malanotteâ A ËǦRRizzoli, G. Pisacane, V. Rupolo, and M. Tsimplis (2006), The Atlantic and Mediterranean Sea as connected systems, in Mediterranean Climate Variability, edited by P. Lionello, P. Malanotteâ A ËǦRRizzoli, and R. Boscolo, pp. 283–322, Elsevier, Oxford, U. K..

REPLY:

Both references have been included in the text.
=====================================================================

Please also note the supplement to this comment:
https://www.ocean-sci-discuss.net/os-2018-143/os-2018-143-AC2-supplement.pdf